# DNN Representations as Codewords: Manipulating Statistical Properties via Penalty Regularization

## Abstract

Performance of Deep Neural Network (DNN) heavily depends on the characteristics of hidden layer representations. Unlike the codewords of channel coding, however, the representations of learning cannot be directly designed or controlled. Therefore, we develop a family of penalty regularizers where each one aims to affect one of the representation's statistical properties such as sparsity, variance, or covariance. The regularizers are extended to perform class-wise regularization, and the extension is found to provide an outstanding shaping capability. A variety of statistical properties are investigated for ten different regularization strategies including dropout and batch normalization, and several interesting findings are reported. Using the family of regularizers, performance improvements are confirmed for MNIST, CIFAR-100, and CIFAR-10 classification problems. But more importantly, our results suggest that understanding how to manipulate statistical properties of representations can be an important step toward understanding DNN, and that the role and effect of DNN regularizers need to be reconsidered.

## 1 Introduction

With a Deep Neural Network (DNN), information contained in the input data $x$ is transformed into multiple *representations* over multiple layers. Performance of machine learning tasks is known to heavily depend on the choice of representations over $p(x)$, but $p(x)$ is almost always unknown and the representations cannot be directly controlled to match an arbitrary design even if $p(x)$ was known. As of today, the best that can be done is to indirectly affect the representations by adding constraints, modifying cost function, and tuning learning process, etc.

In Shannon's information theory, channel coding theory deals with the problem of reliably sending the maximum amount of information through a given channel $p(y|x)$ (Cover & Thomas, 2012). Because the channel is known and fixed, coding becomes a design problem where one needs to design codebook, encoder, and decoder. Usually, the channel is used $n$ times in a sequence to send a codeword of length $n$ (expressed as $x^n$). A codebook is a collection of all codewords that can be chosen and sent to the channel, and each message $w$ with information of interest is mapped into one of the codewords during the design phase.

Because channel coding is a design problem, the optimal solutions are well understood for some of the important applications such as Gaussian channel and Binary Symmetric Channel (BSC). Gaussian channel is the most important continuous alphabet channel problem. It assumes that the received signal $y^n$ is a noisy version of $x^n$, where the noise is independent of $x$ and additive with an i.i.d. Gaussian distribution over the $n$ symbols. Surprisingly, when $n \to \infty$, the optimal codebook turns out to be a collection of codewords that are generated by randomly drawing numbers from a Gaussian distribution. Then, a codeword's $n$ symbols form an i.i.d. Gaussian distribution (for instance, see Chap. 9 of Cover & Thomas (2012)). BSC is one of the most popular discrete alphabet channel problems, and $x$ can take a value of 0 or 1. The received signal $y$ is a corrupted version of $x$ where it is flipped with a fixed probability. For BSC, Hamming code is a well-known solution where redundant bits are included in $x^n$ to resist the corruption (for instance, see Chap. 7 of Cover & Thomas

(2012)). The correction capability is dependent on the minimum Hamming distance among all pairs of two different codewords.

It would be helpful if the elegant channel coding theories can be applied to the design and control of DNN representation, but unfortunately the representation problem is clearly different from the channel coding problem. First of all, a learning problem does not have a fixed and known channel $p(y^n|x^n)$. Secondly, we can design and control the DNN model to use, but we do not have the luxury of explicitly designing codebook. Therefore, the representations can be formed in any way that is possible.

Nonetheless, we can attempt to gain insights and ideas from the established coding theory. In this work, we first recognize that the optimal solution to Gaussian channel problem has i.i.d. Gaussian codewords. Although it is unclear if forcing representations of a layer to have an i.i.d. Gaussian property will be helpful, we experiment the idea by expanding known penalty regularization strategies to include L1, variance, and covariance. L1 and covariance (Cogswell et al., 2016) have been studied before (individually), but with our best knowledge, variance of a unit (neuron) and using a combination of them have not been considered in the literature. Secondly, we recognize that only a single codeword is assigned to a message (label for learning problems) for a well designed codebook. When this idea is applied to learning problems via penalty regularization, the penalty term needs to be applied per-class such that we can shape the codeword of each label. Note that almost all of the existing penalty regularization strategies have been applied to all classes together. Thirdly, we recognize that Gaussian codebook and Hamming codebook are fundamentally different. A Gaussian codebook uses continuous alphabets in an uncorrelated manner over $n$ symbols, but Hamming codebook uses only binary values (0 and 1). With the difference, it is inevitable for Gaussian code to utilize long codewords (very large $n$) and probabilistically guarantee pair-wise distance, while it is essential for Hamming code to utilize carefully designed vector-space structures (orthogonality, null space, etc.) using relatively short codewords. Because we are often interested in a relatively small number of neurons for representations, we consider a regularization strategy where each label's activation for a unit is 'hardened' (by cw-VR regularizer that is introduced later) such that the representation vector is closer to a binary codeword than an i.i.d. Gaussian codeword.

## 1.1 RELATED WORKS

**Regularization**
The classical regularizers apply L2 (Hoerl & Kennard, 1970) and L1 (Tibshirani, 1996) penalties to the *weights* of models, and they are widely used for DNN as well. Wen et al. (2016) extended L1 regularizer by using group lasso to regularize the structures of DNN (i.e., filters, channels, filter shapes, and layer depth). Regularization has been applied to *representations*, too. Srivastava et al. (2014) devised dropout that randomly applies activation masking over the neurons. While dropout is applied in a multiplicative manner, Glorot et al. (2011) used L1 penalty regularization on the activations to encourage sparse representations. XCov proposed by Cheung et al. (2014) minimizes the covariance between autoencoding units and label encoding units of the same layer such that representations can be disentangled. DeCov, developed by Cogswell et al. (2016), is also a penalty regularizer and it minimizes the off-diagonals of a layer's representation covariance matrix. DeCov reduces co-adaptation of units by encouraging units to be decorrelated. It is called CR (Covariance Regularizer) in this study for consistent naming. *Statistics* over mini-batch samples or in-layer activations have been used for regularization, too. Batch normalization proposed by Ioffe & Szegedy (2015) exploits mini-batch statistics to normalize activations. It was developed to accelerate training speed by preventing internal covariate shift, but it was also found to be a useful regularizer. In line with batch normalization, weight normalization, developed by Salimans & Kingma (2016), uses mini-batch statistics to normalize weight vectors. Layer normalization proposed by Ba et al. (2016) is a RNN version of batch normalization, where they compute the mean and variance used for normalization from all of the summed inputs to the neurons in a layer on a single training case. There are many other publications on DNN regularization techniques, but we still do not have a sufficient understanding on how they really work. A recent work by Zhang et al. (2016) shows that the traditional concept of controlling generalization error by regularizing the effective capacity cannot be applied to DNN.

**Class-wise Learning**
True class information is available for supervised learning problems. Traditionally, the class in-

formation has been used only for evaluating the correctness of predictions and the relevant cost function terms. Some of the recent works, however, have adopted the class-wise concept in the learning algorithm itself. In those works, class information is used as a switch or for emphasizing the discriminative aspects over different classes. As an example, Li et al. (2008) proposed a kernel learning method using class-wise information to model the manifold structure. They modify locality preserving projection to be class dependent. Jiang et al. (2011) added label consistent regularizers for learning a discriminative dictionary. As for DNN, a recent work by Liao et al. (2016) used a clustering based regularization that encourages parsimonious representations. In their work, similar representations in sample, spatial, and channel dimensions are clustered and used for regularization such that similar representations are encouraged to become even more similar. While their work can be applied to unsupervised as well as supervised problems, our work utilizes a much simpler method of directly using class labels during training to avoid k-means like clustering. Another recent work by Belharbi et al. (2017) directly uses class labels to encourage similar representations per class as in our work. Their work, however, is based on sum of pair-wise distances among the mini-batch samples of the same labels, and therefore computationally more demanding. The cw-VR (class-wise Variance Regularizer) and cw-CR (class-wise Covariance Regularizer) in this work are very simple penalty regularizers that were designed for the purpose of controlling statistical properties of representations.

## 2 THREE STATISTICAL PROPERTIES AND CLASS-WISE REGULARIZATION

For channel coding problems, we can characterize the statistical properties of optimal codewords as discussed in Section 1. Our goal is to make DNN representation vectors to have such statistical properties and analyze their effects. Because an explicit design and control of representation vector is not possible for the learning problems, we utilize penalty regularizers to manipulate the statistical properties instead.

### 2.1 THREE STATISTICAL PROPERTIES

Three of the most basic statistical properties are considered in this work - sparsity, variance, and covariance. Sparsity over layer $l$'s representation vector $\mathbf{h}_l$ has been extensively studied in the literature. For variance, we are referring to the variance of a unit's activation values over mini-batch samples. When the variance is forced to be very small, the activation value needs to be close to the sample mean for all labels, and therefore the unit loses its discriminative power over multiple labels. While this is undesirable, regularizing variance turns out to be meaningful because the cross-entropy cost function prevents the variance becoming zero, and a healthy compromise can be achieved between cross-entropy and variance terms. This is similar to the situation of classic weight regularization, where the weights actually never become zero by regularization. For covariance, we calculate pair-wise covariance over the unit activations of a layer. When covariance is evaluated to be large for a pair of units (neurons) in the same layer, it indicates that the two are strongly correlated. This is undesirable if we are pursuing i.i.d. property over unit activations, and having a regularizer to control the level of correlation can be useful.

### 2.2 CLASS-WISE REGULARIZATION

To pursue statistical properties for each class, we adopt the concept of class-wise learning.

For instance, it is undesirable if the variance becomes exactly zero for a unit's activation as mentioned above. Variance of zero for a class, however, can be desirable because it simply states that a consistent activation value will be observed over all samples with the same class label. Note that overall variance over all labels can be still large while class-wise variance is zero - as long as inter-class difference exists, the overall variance will not be zero. We combine this concept of class-wise regularization to the three concepts of sparsity, variance, and covariance. Analytical formulations can be found in the following section.

## 3 PENALTY LOSS FUNCTIONS

In this section, we provide the model for calculating basic statistics and formulate the penalty loss functions that are used for regularization.

### 3.1 BASIC STATISTICS

For layer $l$, the output activation vector of a linear filter followed by ReLU is defined as $\mathbf{h}_l = \max(\mathbf{W}_l^\top \mathbf{h}_{l-1} + \mathbf{b}_l, 0)$. Because we will be focusing on layer $l$ for most of the explanations, we drop the layer index and $\mathbf{h}$ is used to indicate $\mathbf{h}_l$ instead. Then, $h_i$ is the $i$th element of $\mathbf{h}$ (i.e. activation of $i$th unit), and $w_{ki}$ is the $(k, i)$ element of $\mathbf{W}$.

To use statistical properties of representations, we define mean of unit $i$, $\mu_i$, and covariance between unit $i$ and unit $j$, $c_{i,j}$, using the $N$ samples in each mini-batch.

$$\mu_i = \frac{1}{N} \sum_n h_{i,n} \tag{1}$$

$$c_{i,j} = \frac{1}{N} \sum_n (h_{i,n} - \mu_i)(h_{j,n} - \mu_j) \tag{2}$$

Here, $h_{i,n}$ is the activation of unit $i$ for $n$th sample in the mini-batch. From equation (2), variance of $i$ unit can be written as below.

$$v_i = c_{i,i} \tag{3}$$

When class-wise statistics need to be considered, we choose a single label $m$ and evaluate mean, covariance, and variance using only the data samples with label $m$ in the mini-batch.

$$\mu_i^m = \frac{1}{|S_m|} \sum_{n \in S_m} h_{i,n} \tag{4}$$

$$c_{i,j}^m = \frac{1}{|S_m|} \sum_{n \in S_m} (h_{i,n} - \mu_i^m)(h_{j,n} - \mu_j^m) \tag{5}$$

$$v_i^m = c_{i,i}^m \tag{6}$$

Here, $S_m$ is the set containing indexes of the samples whose label is $m$, and $|S_m|$ is the cardinality of the set $S_m$.

### 3.2 PENALTY LOSS FUNCTIONS

Using the notations in Section 3.1, the loss functions and their derivatives can be derived and summarized as in Table 1. L1-weight and L2-weight are well-known, and they impose L1 and L2 penalties on the weights, respectively. The rest in the table apply penalties on the representation. L1-rep is similar to L1-weight, but the penalty is applied to the representation $\mathbf{h}$. Obviously, L2 can also be applied to the representation, but it is excluded in this study because it tends to perform worse than L1 when applied to representation. VR (Variance Regularization) calculates variance of each unit's activation over mini-batch dataset and uses the calculated value as the penalty. CR (Cross-covariance Regularization) uses off-diagonal terms of the mini-batch covariance matrix of activations as the penalty term. As mentioned earlier, CR in this work is the same as DeCov presented by Cogswell et al. (2016), but we use the term CR for the consistency of naming. As in DeCov, we subtract variance terms and consider cross-covariance terms only (see penalty loss function in Table 1). cw-VR and cw-CR are similar to VR and CR, respectively, except that the values are calculated for each class using the mini-batch samples with the same class label. cw-L1-rep can be defined, but its penalty loss function turns out to be the same as L1-rep's loss function. Therefore, cw-L1-rep is excluded in this study.

**Interpretation of derivatives**
While the penalty functions were chosen from the three distinct statistical properties and class-wise concept, their derivatives show that some of them are closely related. For the derivatives of VR and

Table 1: Penalty loss functions of regularizers

| | Penalty loss function | Derivatives |
|---|---|---|
| $\Omega_{L1\text{-}weight}$ | $= \sum_k \sum_i |w_{ki}|$ | $\dfrac{\partial \Omega_{L1\text{-}weight}}{\partial w_{ki}} = sign(w_{ki})$ |
| $\Omega_{L2\text{-}weight}$ | $= \sum_k \sum_i w_{ki}^2$ | $\dfrac{\partial \Omega_{L2\text{-}weight}}{\partial w_{ki}} = 2w_{ki}$ |
| $\Omega_{L1\text{-}rep}$ | $= \sum_n \sum_i |h_{i,n}|$ | $\dfrac{\partial \Omega_{L1\text{-}rep}}{\partial h_{i,n}} = sign(h_{i,n})$ |
| $\Omega_{VR}$ | $= \sum_i v_i$ | $\dfrac{\partial \Omega_{VR}}{\partial h_{i,n}} = \dfrac{2}{N}(h_{i,n} - \mu_i)$ |
| $\Omega_{CR}$ | $= \sum_i \sum_j (c_{i,j})^2 - \sum_i (v_i)^2$ | $\dfrac{\partial \Omega_{CR}}{\partial h_{i,n}} = \dfrac{2}{N} \sum_{j \neq i} c_{i,j}(h_{j,n} - \mu_j)$ |
| $\Omega_{cw\text{-}VR}$ | $= \sum_m \sum_i v_i^m$ | $\dfrac{\partial \Omega_{cw\text{-}VR}}{\partial h_{i,n}} = \dfrac{2}{|S_m|}(h_{i,n} - \mu_i^m), n \in S_m$ |
| $\Omega_{cw\text{-}CR}$ | $= \sum_m (\sum_i \sum_j (c_{i,j})^2 - \sum_i (v_i)^2)$ | $\dfrac{\partial \Omega_{cw\text{-}CR}}{\partial h_{i,n}} = \dfrac{2}{|S_m|} \sum_{j \neq i} c_{i,j}^m(h_{j,n} - \mu_j^m), n \in S_m$ |

CR, it can be observed that they have similar structures. If VR's derivative $\dfrac{\partial \Omega_{VR}}{\partial h_{i,n}}$ becomes zero for all $i$, then CR's derivative $\dfrac{\partial \Omega_{CR}}{\partial h_{i,n}}$ becomes zero as well. The vice versa does not hold, but the effects of VR and CR can be expected to be similar or at least related to each other for the learning process. In the same way, the relationship between cw-VR and cw-CR is the same as the relationship between VR and CR. Therefore, we can expect cw-VR and cw-CR to have similar effects, too. On the other hand, the derivative of L1-rep has a distinct formulation, and it can be expected to have a distinct effect on learning.

There is another important effect that is not necessarily obvious from the derivative formulations. For L1-weight and L2-weight, the derivatives are dependent on the weights $w_{ki}$ only, and they are independent of the activations $h_{i,n}$. Therefore, the weights need to become smaller to reduce the regularization penalty. For the other five representation regularizers, their derivatives are all dependent on activation $h_{i,n}$. So, a simple way to reduce the regularization penalties is to scale the activations to small values (instead of satisfying the balances among the terms in the equation to reach zero gradients and force the desired statistical properties). This scaling will not have any effect on prediction output as long as all the elements of $\mathbf{h}^l$ are scaled together to $\alpha \mathbf{h}^l$ - the last softmax layer works as a normalization function for the output layer, and therefore the cross-entropy penalty term is not affected by such a scaling. This means that there is a chance for the learning algorithm to squash activations just so that representation regularization terms can be ignored. As we will see later, indeed activation squashing happens by learning, but the desired statistical properties are still sufficiently enforced. Nonetheless, it must be possible to design better penalty regularizers that are immune to activation squashing, and such regularizers might be much more effective for manipulating statistical properties of representations.

## 4 EXPERIMENTS - MNIST

In this section, we consider ten regularization strategies and compare them using the MNIST dataset (LeCun et al., 1998). We use a Multilayer Perceptron (MLP) with five hidden fully connected layers and an output layer. Each hidden layer has 100 units with Rectified Linear Unit (ReLU) activation function, and the output layer consists of 10 softmax units. All experiments (in this work) were carried out using TensorFlow 1.3.

Table 2: Error performance of popular regularizers (MNIST)

| Layer | Baseline | Penalty on weight | | Implicit method | |
|---|---|---|---|---|---|
| | | L1-weight | L2-weight | Dropout | BN |
| All | **3.06**±0.15 | **2.90**±0.08 | **2.96**±0.09 | **4.08**±0.06 | **2.69**±0.06 |

Table 3: Error performance of representation regularizers (MNIST)

| Layer | All classes | | | Class-wise | |
|---|---|---|---|---|---|
| | L1-rep | VR | CR | cw-VR | cw-CR |
| Output | **2.61**±0.04 | **2.67**±0.15 | **2.62**±0.07 | **2.56**±0.02 | **2.55**±0.08 |
| Layer 5 | **2.61**±0.11 | 2.70±0.03 | 2.67±0.04 | 2.63±0.05 | 2.61±0.06 |
| Layer 4 | 2.75±0.05 | 2.89±0.11 | 2.69±0.13 | 2.67±0.12 | 2.71±0.04 |
| Layer 3 | 3.35±0.08 | 3.16±0.09 | 3.11±0.13 | 3.22±0.06 | 3.22±0.06 |
| Layer 2 | 3.40±0.11 | 3.15±0.21 | 3.01±0.10 | 3.14±0.10 | 3.24±0.11 |
| Layer 1 | 4.31±0.14 | 2.98±0.09 | 3.13±0.09 | 3.25±0.04 | 3.14±0.03 |

Table 4: Error performance of representation regularizers - multiple layers (MNIST)

| | L1-rep | VR | CR | cw-VR | cw-CR |
|---|---|---|---|---|---|
| Output | 2.61±0.04 | 2.67±0.15 | 2.62±0.07 | 2.56±0.02 | 2.55±0.08 |
| Output, 5 | **2.48**±0.12 | 2.67±0.11 | **2.43**±0.08 | **2.46**±0.07 | 2.55±0.10 |
| Output, 5, 4 | 2.78±0.11 | **2.58**±0.06 | 2.80±0.12 | 2.53±0.07 | **2.48**±0.07 |
| Output, 5, 4, 3 | 2.79±0.10 | 2.78±0.08 | 2.83±0.14 | 2.80±0.10 | 2.72±0.04 |
| Output, 5, 4, 3, 2 | 3.19±0.10 | 2.91±0.13 | 2.77±0.07 | 2.90±0.10 | 2.75±0.07 |
| All | 3.26±0.09 | 2.86±0.07 | 2.80±0.08 | 2.83±0.07 | 2.85±0.12 |

## 4.1 PERFORMANCE RESULTS

For each regularization term, the level of regularization was determined by tuning the penalty loss weight using a validation dataset and a grid search. Then, we trained each model five-times and calculated the test error performance as the average and one standard deviation over the five performance results. In Table 2 and Table 3, the results show that representation regularizers outperform the popular regularizers and that the representation strategies perform better when applied to upper layers of DNN. Interestingly, the best performance is achieved by applying representation regularization to the output layer as shown in Table 3. This might be because the regularizer directly affects only the regularizing layer and the layers below, or because manipulating statistical properties is more effective for the higher layer representations that have stronger or codeword-like structures. To better understand the effect of a layer, multiple layer results are shown in Table 4. The best performance is achieved when output layer is regularized together with one or two upper hidden layers. Among all the results in the three tables, CR performs best and achieves 2.43% of error.

## 4.2 STATISTICAL PROPERTIES OF 10 REGULARIZATION STRATEGIES

We use nine metrics to compare the statistical properties of the ten regularization strategies. Among the nine metrics, first seven of them are calculated by directly evaluating the penalty loss functions shown in Table 1. The raw evaluation values, however, are difficult to interpret because they have different scales. So, we normalize the metrics as following (see the raw evaluation values shown in Table 10 and Table 11). First, square-root is applied to L2-weight, VR, and CR because their units are quadratic, and square-root of square-root is applied to cw-VR and cw-CR because their units are quartic. Then, all the metrics of each regularizer are divided by the regularizer's own $\sqrt{\Omega_{L2\text{-}weight}}$ such that all are normalized with respect to its 2-norm weight values. Finally, all the metrics are normalized by baseline's metrics and 100 is multiplied such that we can focus on the relative change in percentage compared to the baseline's metrics. The remaining two metrics are the average number of activated classes per unit as the measure of sparsity and ratio of dead units, and they are explained in Appendix C. $\Omega_{L1\text{-}weight}$ and $\Omega_{L2\text{-}weight}$ are calculated from the weights of all layers excluding biases, and the others are calculated from layer 5's activations using test dataset.

Table 5: Evaluation of statistical properties (layer 5) - popular strategies

| Metric | Baseline | Penalty on weight | | Implicit method | |
|---|---|---|---|---|---|
| | | L1-weight | L2-weight | Dropout | BN |
| $\Omega_{L1-weight}$ (all) | 100.00 | 88.05 | 92.25 | 99.14 | 84.82 |
| $\Omega_{L2-weight}$ (all) | 100.00 | 100.00 | 100.00 | 100.00 | 100.00 |
| $\Omega_{L1-rep}$ | 100.00 | 115.28 | 110.26 | 36.11 | 16.94 |
| $\Omega_{VR}$ | 100.00 | 113.97 | 109.58 | 61.18 | 27.69 |
| $\Omega_{CR}$ | 100.00 | 111.55 | 107.51 | 39.35 | 5.80 |
| $\Omega_{cw-VR}$ | 100.00 | 114.08 | 109.68 | 72.91 | 50.50 |
| $\Omega_{cw-CR}$ | 100.00 | 112.68 | 108.55 | 78.49 | 20.54 |
| $Avg\_Act\_Class$ | 5.24 | 5.54 | 5.35 | 4.60 | 2.48 |
| $Ratio\_Dead\_Unit$ | 14% | 5% | 9% | 0% | 1% |

Table 6: Evaluation of statistical properties (layer 5) - representation regularizers

| Metric | All classes | | | Class-wise | |
|---|---|---|---|---|---|
| | L1-rep | VR | CR | cw-VR | cw-CR |
| $\Omega_{L1-weight}$ (all) | 93.08 | 96.42 | 95.83 | 86.85 | 84.14 |
| $\Omega_{L2-weight}$ (all) | 100.00 | 100.00 | 100.00 | 100.00 | 100.00 |
| $\Omega_{L1-rep}$ | **1.07** | 9.16 | 9.73 | 3.41 | 5.49 |
| $\Omega_{VR}$ | 7.77 | 9.24 | 9.42 | **3.91** | 5.28 |
| $\Omega_{CR}$ | 0.33 | 0.64 | 0.63 | **0.15** | 0.27 |
| $\Omega_{cw-VR}$ | 19.85 | 28.12 | 29.61 | **11.25** | 14.27 |
| $\Omega_{cw-CR}$ | 3.69 | 6.79 | 7.15 | **1.66** | 2.37 |
| $Avg\_Act\_Class$ | **0.23** | 5.12 | 5.38 | 4.14 | 5.29 |
| $Ratio\_Dead\_Unit$ | **77%** | 9% | 5% | 23% | 7% |

We can observe two distinct groups of regularizers by investigating Table 5 and Table 6. We can observe that the representation regularizers have much smaller values for the representation metrics. This is because the representation regularizers squash activations in the way described in Section 3.2. As mentioned in Section 3.2, VR and cw-VR are related to CR and cw-CR, respectively. We can see that their values of metrics are similar to each other. Despite this similarity of the five representation regularization, L1-rep and cw-VR have unique characteristics. L1-rep obviously enforces sparsity and causes much more dead units than the others. The regularizer cw-VR always shows the smallest metric values among four strategies (VR, CR, cw-VR, and cw-CR). This can be an evidence of the four regularizers' close relationship. The metric values of dropout and batch normalization (BN) are located somewhere between baseline and representation regularizers. They cause similar effects on representation metrics as the representation regularizers, but much less effect are observed. It is also interesting to note that both dropout and BN have only 0~1% of dead units (neurons). Dropout and BN are implicit methods in the sense that they do not target any particular statistical property, but they certainly seem to have distinct effects compared to the other regularizers.

## 4.3 VISUALIZATION OF REPRESENTATIONS

Due to activation squashing, metrics of statistical properties can become misleading. Therefore, we visualize the representation of Layer 5 to more intuitively understand the statistical properties that are affected by the regularizers. Samples for three regularizers are shown in Figure 1 and Figure 2, and all figures for the ten regularizers are shown in appendix (Figure 3 and Figure 4).

**Histogram of a single unit**
We first visualize the distribution of activation per unit in Figure 1 to observe sparsity and variance properties. Activation histograms were generated by using 10,000 test data, and each color corresponds to a different class. Since activations are generated as the output of ReLU activation function, many have zero value that can distort the histogram. We, therefore, excluded zeros from activations when drawing the histogram plots. In Figure 1(a), it can be seen that baseline has a large class-wise variance and inter-class overlaps. The histogram of cw-VR in (b), however, shows the effect of separating the classes because class-wise variance is significantly reduced. For each class,

the activation is 'hardened'. L1-rep in (c) can be confirmed to have only one class that is activated, and this confirms the sparsity. As described in Table 6, $Avg\_Act\_Class$ of L1-rep is close to zero, so most of its histograms show very few active samples.

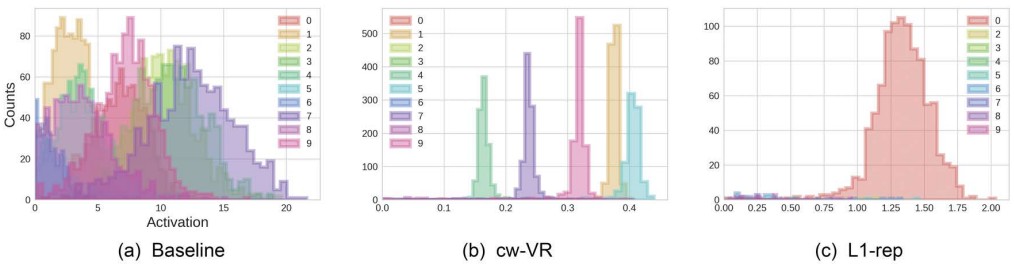

(a) Baseline          (b) cw-VR          (c) L1-rep

Figure 1: Histogram of a sample nueron's activation values over test dataset. The sample was chosen from $\mathbf{h}_5$. Compared to baseline, cw-VR clearly shows non-overlapping distributions for different labels. L1-rep shows a similar distribution shape as in the baseline, but only a single label is activated in this example. Best viewed in color.

**Scatter plot of a pair of units**

To show the relationship between two representation units, we randomly chose two units from a representation vector $\mathbf{h}_5$ and drew a scatter plot of their activation values for the test dataset. As shown in Figure 2, baseline shows a modest linearity, which is consistent with the high covariance value. Since CR in (b) reduces cross-covariance per unit, it can be seen that overall linearity is significantly reduced compared to the baseline and the randomly chosen pair of units becomes almost independent. In the same way, cw-CR has reduced class-wise cross-covariance. Furthermore, its class-wise variance is small and thus end up having small ball-shaped concentrations of points.

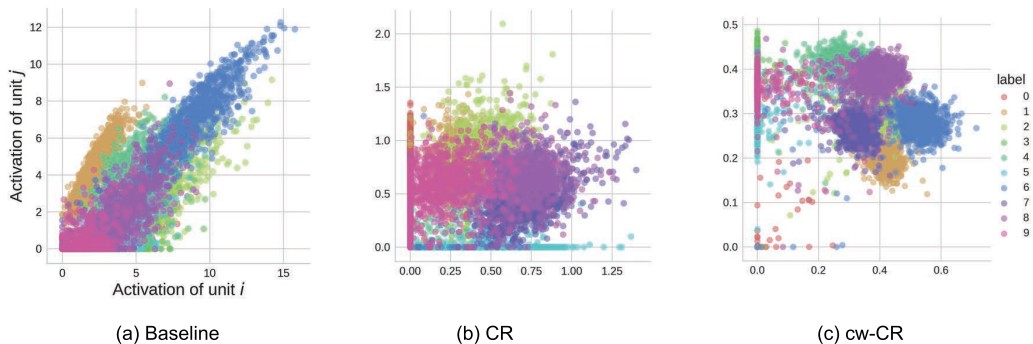

(a) Baseline          (b) CR          (c) cw-CR

Figure 2: Scatter plot of activation values of randomly chosen two units from $\mathbf{h}_5$. Compared to baseline, CR has clearly less correlation indicating less co-adaptation. cw-CR also shows low co-adaptation, but it has smaller ball shapes per label because of the low class-wise variance. Best viewed in color.

## 5 EXPERIMENTS - CIFAR-10/100

While performance improvement is not the primary focus of this work, we provide additional test results with performance evaluations and show that the representation regularizers are useful for pushing the accuracy performance to the next level. In particular, we provide additional test results for CIFAR-100 and CIFAR-10 datasets (Krizhevsky & Hinton, 2009). For CIFAR-100, we have chosen a toy CNN architecture to confirm performance improvement of representation regularizers. Concurrently using two of the regularizers is experimented as well. For CIFAR-10, we have

Table 7: Error performance of regularizers (CIFAR-100)

| Regularizer | | Train error | Test error |
|---|---|---|---|
| Baseline | None | 25.50 | **56.02** |
| Penalty on weight | L1-weight | 18.16 | 55.99 |
| | L2-weight | 33.75 | 54.93 |
| Implicit method | Dropout (fc) | 28.02 | 55.28 |
| | Dropout (all) | 79.28 | 80.08 |
| | BN (fc) | 28.28 | 55.33 |
| | BN (all) | 8.63 | 57.82 |
| Penalty on representation | Single | | |
| | L1-rep | *98.93* | *99.00* |
| | VR | 27.02 | **53.66** |
| | CR | 33.24 | 54.67 |
| | cw-VR | 22.85 | 54.15 |
| | cw-CR | 27.84 | 53.78 |
| Combination | VR + CR | 13.88 | 54.68 |
| | VR + cw-VR | 19.43 | 56.12 |
| | VR + cw-CR | 28.53 | 54.94 |
| | CR + cw-VR | 21.11 | **53.30** |
| | CR + cw-CR | 18.05 | 54.75 |
| | cw-VR + cw-CR | 25.77 | 55.64 |
| | L1-rep + VR | *98.93* | *99.00* |
| | L1-rep + CR | *98.93* | *99.00* |
| | L1-rep + cw-VR | *98.93* | *99.00* |
| | L1-rep + cw-CR | *98.93* | *99.00* |

tested representation regularizers using Residual Network (ResNet) that is known as one of the best performing deep neural networks for image data.

## 5.1 COMBINING MULTIPLE STRATEGIES: CIFAR-100

We use a toy CNN network for experimenting with CIFAR-100. The CNN network consists of four convolution layers and a fully connected layer, all with 100 hidden units. ReLU is used as the activation function. The second, third, and fourth convolution layers are followed by a max pooling layer. The last 10,000 instances of 50,000 training data were used as the validation data. Using the validation data, validation performance was evaluated for the regularizer weight values of {0.1, 0.01, 0.001, 0.0001}. The best weight value was found for each regularizer, and the test performance was evaluated for the fixed weight values. For representation regularizers, regularization was applied to the fully connected layer. The performance results are shown in Table 7. From the table, it can be seen that the test error is improved from baseline 56.02% to 53.66% by using a single regularizer (VR) and to 53.30% by using two regularizers (CR and cw-VR). Therefore, 2.72% of improvement is achieved by the best performing regularizer combination. Aside from the performance improvement, it is interesting to observe that L1-rep consistently fails to train for the CIFAR-100 data. With 100 labels, too much sparsity might hurt the performance. This is a plausible hypothesis considering that we have only 100 neurons to encode 100 labels. A shared use of neurons over multiple classes might be a better direction to pursue. In general, the relationship between the number of labels and the desired statistical properties of representation remains a topic to be studied.

## 5.2 PERFORMANCE IMPROVEMENT OF RESNET-32

ResNet was first proposed by He et al. (2016). ResNet consists of multiple basic blocks that are serially connected, and shortcut connections to force residuals to be calculated. We apply five regularization strategies without modifying the ResNet-32 architecture. Regularization was applied to the output layer only. Experimental results in Table 8 show that performance is improved over the state-of-the-art ResNet-32 model, and cw-VR shows the best performance. This indicates that representation regularizers are compatible with ResNet, and most likely also with other state-of-the-art models.

Table 8: Error performance of regularizers on ResNet-32 (CIFAR-10)

| Model | He et al. | Ours |
|---|---|---|
| ResNet-32 | **7.51** | **7.39** |
| ResNet-32 + L1-rep | | 7.27 |
| ResNet-32 + VR | | 7.22 |
| ResNet-32 + CR | | 7.27 |
| ResNet-32 + cw-VR | | **7.17** |
| ResNet-32 + cw-CR | | 7.21 |

## 6 CONCLUSION

In this work, we have investigated five different penalty regularizers for manipulating statistical properties of DNN representations. The regularizers were conceived by examining optimal code-words of well-known channel coding problems, and the three statistical properties of sparsity, variance, and covariance were integrated into the regularizers along with the concept of class-wise regularization. It was found that many statistical properties including cross-covariance, co-adaptation, per-class variance, average number of active class per-unit, and the ratio of dead units can be manipulated. Each regularizer, however, tended to manipulate multiple properties at the same time, making it difficult to manipulate each property individually. While manipulation was shown to be possible and helpful for improving the performance of all three DNN classification problems that were investigated, it is still unclear if any statistical property of representation is generally helpful when strengthened. Due to the complicated nature of learning process where back-propagation affects not only the signal of interest but also other signals and irrelevant noise, it still remains an open question on how to establish procedures that *generally* improve learning of any deep learning problems.

The contributions of this work can be summarized as follow. First, a complete set of very simple regularizers for controlling sparsity, variance, and covariance of representations was presented. Among them, VR, cw-VR, and cw-CR have been designed and used for the first time and they work very well. The visualizations clearly show that the new regularizers are effective for manipulating statistical properties of representations in new ways. Secondly, by analyzing statistical properties in a quantitative way, we have shown that none of the popular regualrizers works in a distinct way. Even the well-known dropout does not control co-adaptation(covariance) only. In fact, sparsity and class-wise variance are affected together by dropout, and therefore it is difficult to claim if indeed reduction in co-adaptation is why dropout works well. Thirdly, we have provided partial results on which statistical properties can be helpful or harmful for different learning tasks (tasks with more labels, with more complexity, etc.). This part needs to be further investigated to see if general rules can be derived.

ACKNOWLEDGMENTS

To be added.

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

APPENDIX

## A PERFORMANCE OF POPULAR REGULARIZERS WHEN APPLIED TO EACH LAYER

Table 9: Error performance of popular regularizers - applied to each layer

| Layer | Baseline | Penalty on weight | | Implicit method | |
|-------|----------|-------------------|-----------|-----------------|-----------|
| | | L1-weight | L2-weight | Dropout | BN |
| All | | **2.90**±0.08 | 2.96±0.09 | 4.08±0.06 | 2.69±0.06 |
| Output | | 3.02±0.15 | 2.96±0.06 | 2.99±0.18 | 2.97±0.08 |
| Layer 5 | | 2.98±0.05 | 2.99±0.13 | 2.80±0.08 | 3.04±0.09 |
| Layer 4 | **3.06**±0.15 | 2.98±0.08 | 2.98±0.09 | **2.67**±0.05 | 2.84±0.15 |
| Layer 3 | | 3.04±0.09 | 3.03±0.18 | **2.67**±0.16 | 2.94±0.13 |
| Layer 2 | | 2.91±0.05 | 2.76±0.16 | 2.70±0.08 | 2.84±0.16 |
| Layer 1 | | 2.93±0.05 | **2.52**±0.10 | 3.07±0.07 | **2.58**±0.07 |

## B EVALUATION OF STATISTICAL PROPERTIES

Table 10: Evaluation of statistical properties (raw) - popular regularizers

| Property | Baseline | Penalty on weight | | Implicit method | |
|----------|----------|-------------------|-----------|-----------------|-----------|
| | | L1-weight | L2-weight | Dropout | BN |
| $\Omega_{L1\text{-}weight}$ (all) | 9795.03 | 7504.60 | 8220.21 | 9461.52 | 9488.60 |
| $\Omega_{L2\text{-}weight}$ (all) | 607.46 | 459.85 | 502.71 | 576.60 | 792.24 |
| $\Omega_{L1\text{-}rep}$ | $3.24 \times 10^6$ | $3.25 \times 10^6$ | $3.25 \times 10^6$ | $1.14 \times 10^6$ | $6.27 \times 10^5$ |
| $\Omega_{VR}$ | 865.69 | 851.24 | 860.34 | 307.64 | 86.59 |
| $\Omega_{CR}$ | 58178.00 | 54803.10 | 55650.20 | 8551.84 | 255.31 |
| $\Omega_{cw\text{-}VR}$ | 2398.03 | 2327.79 | 2377.46 | 610.58 | 265.46 |
| $\Omega_{cw\text{-}CR}$ | 63726.20 | 58891.60 | 60610.20 | 21795.60 | 193.26 |

Table 11: Evaluation of statistical properties (raw) - representation regularizers

| Property | All classes | | | Class-wise | |
|----------|-------------|----|----|------------|-------|
| | L1-rep | VR | CR | cw-VR | cw-CR |
| $\Omega_{L1\text{-}weight}$ (all) | 9975.62 | 9732.16 | 9826.06 | 9843.84 | 9772.89 |
| $\Omega_{L2\text{-}weight}$ (all) | 727.22 | 645.03 | 665.57 | 813.20 | 853.98 |
| $\Omega_{L1\text{-}rep}$ | 38183.20 | $3.06 \times 10^5$ | $3.39 \times 10^5$ | $1.28 \times 10^5$ | $2.11 \times 10^5$ |
| $\Omega_{VR}$ | 6.26 | 7.85 | 8.43 | 1.78 | 3.40 |
| $\Omega_{CR}$ | 0.80 | 2.55 | 2.55 | 0.19 | 0.64 |
| $\Omega_{cw\text{-}VR}$ | 5.34 | 16.91 | 22.15 | 0.69 | 1.97 |
| $\Omega_{cw\text{-}CR}$ | 0.17 | 1.53 | 2.00 | $8.83 \times 10^{-3}$ | 0.04 |

## C METRICS

**Activated class and dead unit**
ReLU's output becomes positive when the input has a positive value. In this work, we say a class is activated for a nueron if the probability of the neuron's output being positive is above a threshold for the given class. We use the entire test dataset to check the probability, and threshold value of 0.9 is used for the evaluations. If many classes are activated for a neuron, it indicates that the neuron is used for representations of many classes. On the other hand, if only a single class is activated for a neuron, it indicates that the neuron is used for representations of only one class and kept zero for all the other classes. When the number of activated class is zero for a neuron, it indicates that the neuron does not carry any information and may be ignored. Such a neuron is called a dead unit. The equations below show how to calculate if a class $m$ is activated for a nueron $i$. $I$ is an indicator function, and $N_u$ is the number of units in the layer.

$$Num\_Act\_InClass(i, m) = \sum_{n \in S_m} I(h_{i,n} > 0)$$

$$Act\_Class(i, m) = I(\frac{Num\_Act(i, m)}{|S_m|} > threshold)$$

**Average number of activated classes**
The number of activated classes can be calculated for each unit. Then, the average number of activated classes can be calculated over all units in the same layer. When $Avg\_Act\_Class$ is large for a regularizer, it means the regularizer tends to encourage many units to be used for representations. If the value is small, it indicates the regularizer makes only a small number of units to be coded in positive values for the representation.

$$Num\_Act\_Class(i) = \sum_{m} Act\_Class(i, m)$$

$$Avg\_Act\_Class = \frac{\sum_{i} Num\_Act\_Class(i)}{N_u}$$

**Ratio of dead units**
Typically, 'dead neuron' is widely used to represent neurons that are not activated - output is zero all the time over all classes. To extend the concept of 'activated class', we define $All\_Class\_Dead(i)$ and $Ratio\_Dead\_Unit$ as below. When $Ratio\_Dead\_Unit$ is large, it indicates many of the neurons can be removed without affecting the representation.

$$All\_Class\_Dead(i) = I(\sum_{m} Act\_Class(i, m) = 0)$$

$$Ratio\_Dead\_Unit = \frac{\sum_{i} All\_Class\_Dead(i)}{N_u}$$

# D    VISUALIZATION OF REPRESENTATIONS

## D.1    REPRESENTATION HISTOGRAMS OF 10 REGULARIZERS

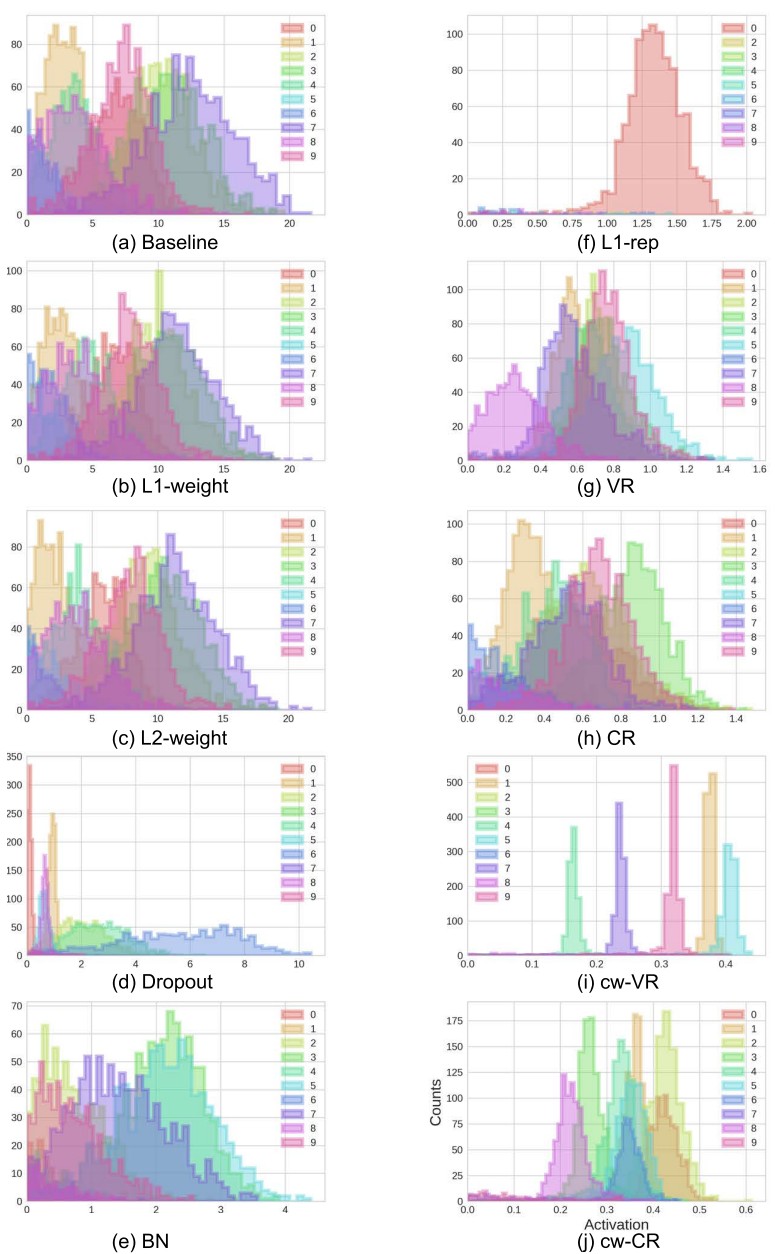

Figure 3: Histograms of activation values for 10 regularizers. Best viewed in color.

D.2    REPRESENTATION SCATTER PLOTS OF 10 REGULARIZATION STRATEGIES

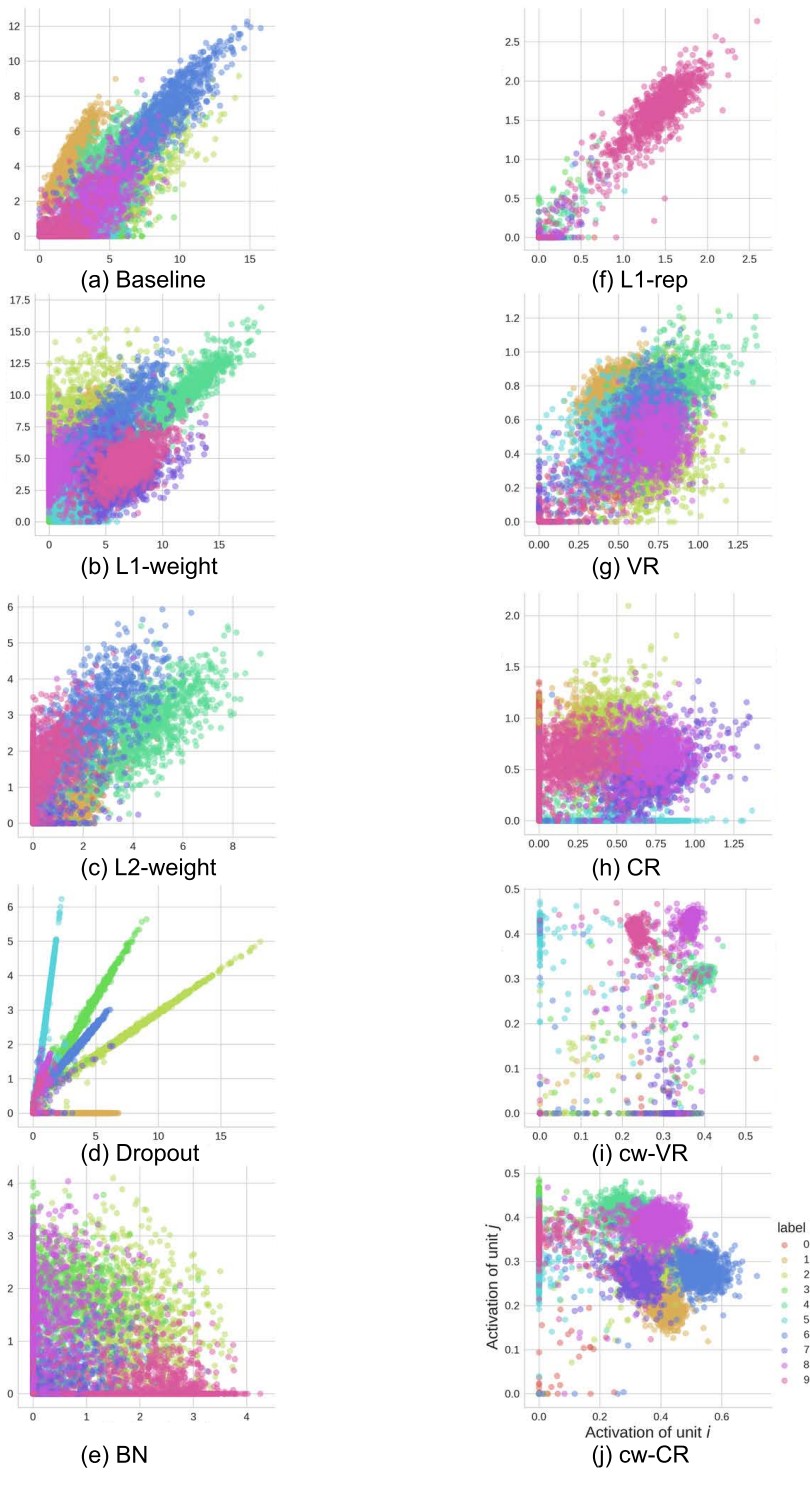

Figure 4: Scatter plots of activation values of two units (neurons) for 10 regularizers. Best viewed in color.

