# OpenReview forum: "DNN Representations as Codewords: Manipulating Statistical Properties via Penalty Regularization"
_ICLR.cc/2018/Conference — Reject_

### Official Review · AnonReviewer1 · 2017-11-27
**This paper presents a set of regularizers which aims for manipulating the statistical properties like sparsity, variance and covariance.**

**Rating:** 5
**Confidence:** 5

**Review:**

This paper presents a set of regularizers which aims for manipulating the statistical properties like sparsity, variance and covariance. While some of the proposed regularizers are applied to weights, most are applied to hidden representations of neural networks. Class-wise regularizations are also investigated for the purpose of fine-grained control of statistics within each class. Experiments over MNIST, CIFAR10 and CIFAR100 demonstrate the usefulness of this technique.

The following related work also studied the regularizations on hidden representations which are motivated from clustering perspective and share some similarities with the proposed one. It would be great to discuss the relationship.

Liao, R., Schwing, A., Zemel, R. and Urtasun, R., 2016. Learning deep parsimonious representations. NIPS.

Pros:
(1) The paper is clearly written.

(2) The visualizations of hidden activations are very helpful in understanding the effect of different regularizers.

(3) The proposed regularizations are simple and computationally efficient.

Cons:
(1) The novelty of the paper is limited as most of the proposed regularizers are more or less straightforward modifications over DeCov.

(2) When we manipulate the statistics of representations we aim for something, like improving generalization, interpretability. But as pointed out by authors, improvement of generalization performance is not the main focus. I also do not find significant improvement from all experiments. Then the question is what is the main benefit of manipulating various statistics?

I have an additional question as below:
In measuring the ratio of dead units, I notice authors using the criterion of “not activated on all classes”. However, do you check this criterion over the whole epoch or just some mini-batches?

Overall, I think the paper is technically sound. But the novelty and significance are a bit unsatisfactory. I would like to hear authors’ feedback on the issues I raised.

---

> ### Author Response · Authors · 2017-12-28
> **Response:**
>
> Thanks for your feedback. As you can see from the above responses to the other two reviewers, we fully agree with the feedbacks from the three reviewers.
>
> Thank you for your comment on the visualizations. Actually, we were originally investigating mutual information related problems but have noticed a few unexpected shapes while performing visualizations. The observation has lead us to the idea of manipulating statistical properties of representations as summarized in this work. When this work is viewed as an effort to introduce a few new regularizers, indeed our new regularizers are just extensions of the existing ideas (well, but not trivial at all...). Rather, our intention was to investigate how statistical properties of representations are generally related to the learning. Our contributions were not clear in the original writing, and therefore we have added a paragraph at the end of conclusion to summarize the contributions. For your convenience, they are repeated below.
>
> *** The contributions of this work can be summarized as follow. First, a complete set of very simple regularizers for controlling sparsity, variance, and covariance of representations was presented. Among them, VR, cw-VR, and cw-CR have been designed and used for the first time and they work very well. The visualizations clearly show that the new regularizers are effective for manipulating statistical properties of representations in new ways. Secondly, by analyzing statistical properties in a quantitative way, we have shown that none of the popular regualrizers works in a distinct way. Even the well-known dropout does not control co-adaptation(covariance) only. In fact, sparsity and class-wise variance are affected together by dropout, and therefore it is difficult to claim if indeed reduction in co-adaptation is why dropout works well. Thirdly, we have provided partial results on which statistical properties can be helpful or harmful for different learning tasks (tasks with more labels, with more complexity, etc.). This part needs to be further investigated to see if general rules can be derived. ***
>
> We regret that we have submitted a bit premature result, and we will work on a better version of this study. In any case, we truly appreciate your helpful and thoughtful comments.
>
> >> In measuring the ratio of dead units, I notice authors using the criterion of “not activated on all classes”. However, do you check this criterion over the whole epoch or just some mini-batches?
> ==> The dead units were checked by using the entire test data set. The equations for deriving them can be found in Appendix C, and the writing in there was revised to improve its readability.

---

### Official Review · AnonReviewer3 · 2017-11-27
**Paper proposed new regulariziation schemes**

**Rating:** 5
**Confidence:** 3

**Review:**

1. Summary
The authors of the paper compare the learning of representations in DNNs with Shannons channel coding theory, which deals with reliably sending information through channels. In channel coding theory the statistical properties of the coding of the information can be designed to fit the task at hand. With DNNs the representations cannot be designed in the same way. But the representations, learned by DNNs, can be affected indirectly by applying regularization. Regularizers can be designed to affect statistical properties of the representations, such as sparsity, variance, or covariance. The paper extends the regularizers to perform per-class regularization. This makes sense, because, for example, forcing the variance of a representation to go towards zero is undesirable as it would state that the unit always has the same output no matter the input. On the other hand having zero variance for a class is desirable as it means that the unit has a consistent activation for all samples of the same class. The paper compares different regularization techniques regarding their error performance. They find that applying representation regularization outperforms classical approaches such as L1 and L2 weight regularization. They also find, that performing representation regularization on the last layer achieves the best performance. Class-wise methods generally outperform methods that apply regularization on all classes.

2. Remarks
Shannons channel coding theory was used by the authors to derive regularizers, that manipulate certain statistical properties of representations learned by DNNs. In the reviewers opinion, there is no theoretical connection between DNNs and channel theory. For one, DNNs are no channels in the sense that they transmit information. DNNs are rather pipes that transform information from one domain to another, where representations are learned as an intermediate model as the information is being transformed. Noise introduced in the process is not due to a faulty channel but due to the quality of the learned representations themselves. The paper falls short in explaining how DNNs and Shannons channel coding theory fit together theoretically and how they used it to derive the proposed regularizers. Despite the theoretical gap between the two was not properly bridged by the authors, channel coding theory is still a good metaphor for what they were trying to achieve.
The authors recognize that there is similar research being done independently by Belharbi et al. (2017). The similarities and differences between the proposed work and Belharbi et al. should be discussed in more detail.
The authors conclude that it is unclear which statistical properties of representations are generally helpful when being strengthened. It would be nice if they had derived at least a set of rules of thumb. Especially because none of the regularizers described in the paper only target one specific statistical property but multiple. One good example that was provided, is that L1-rep consistently failed to train on CIFAR-100, because too much sparsity can hurt performance, when having many different classes (100 in this case). These kinds of conclusions will make it easier to transfer the presented theory into practice.

3. Conclusion
The comparison between DNNs and Shannons channel coding theory stands on shaky ground. The proposed regularizes are rather simple, but perform well in the experiments. The effect of each regularizer on the statistical properties of the representation and the relations to previous work (especially Belharbi et al. (2017)) should be discussed in more detail.

---

> ### Author Response · Authors · 2017-12-28
> **Response:**
>
> Thanks for your feedback. We fully agree that DNN is different from channel coding problems, and a few reasons can be found in the original writing of the introduction section. Furthermore, we agree that our paper lacks enough analysis that can connect between coding theories and DNN (please see the above responses to AnonReviewer2). Nonetheless, our work provides a few contributions including 1. a complete set of very simple regularizers for controlling sparsity, variance, and covariance of representations (VR, cw-CR, and cw-CR have been designed and used for the first time and they work very well; as pointed out by AnonReviewer1, the visualizations show that they are very effective for manipulating statistical properties of representations), 2. observation/analysis for showing that none of the popular regualrizers works in a distinct way (even well-known dropout does not control co-adaptation(covariance) only; in fact, sparsity and class-wise variance are affected together by dropout, and therefore it is impossible to tell if indeed reduction in co-adaptation is why dropout works well), and 3. partial results on which statistical properties can be helpful or harmful for different learning tasks (tasks with more labels, with more complexity, etc.). We agree with your feedback that the contribution #3 should have been further investigated and better summarized. We are currently working on the issue. Overall, we agree with your assessment, and have made an update at the end of conclusion section to clarify the contributions. We will try to have a better version written in the coming few months. :) We truly appreciate your careful review.
>
> Regarding Belharbi et al. (2017), indeed the paper has a few overlappings with our work. The work focuses on reducing variation among same-class representations using a regularizer, and it also investigates layer dependency on applying the regularizer. The findings on layer dependency is consistent with our findings. From the technical perspective, our work is different because our regularizers are much simpler to implement (no sample pair-wise calculation) and because we designed and investigated a set of regularizers instead of a single regularizer. Our focus was to investigate statistical properties of representations and their effects on learning. We have updated our writing to introduce Belharbi et al. (2017) in a proper way, and we have also included Liao et al. (2016) that was pointed out by AnonReviewer1.

---

### Official Review · AnonReviewer2 · 2017-11-30
**DNN Representations as Codewords: Manipulating Statistical Properties via Penalty Regularization**

**Rating:** 5
**Confidence:** 4

**Review:**

This is a well-written paper which starts on a good premise: DNNs learn representations; good representations are good for prediction; they can be seen as codes of the input information, so let us look at coding/communication/information theory for inspiration. This is fine and recent work from N Tishby's group develop some intriguing observations from this. But this paper doesn't follow through the information/communication story in any persuasive way. All that is derived is that it may be a good idea to penalise large variations in the representation -- within-class variations, in particular. The paper does a good job of setting up and comparing empirical performance of various regularizers (penalties on weights and penalties on hidden unit representations) and compares results against a baseline. Error rates (on MNIST, for example) are very small (baseline 3% versus the best in this paper 2.43%), but, if I am right, these are results quoted on a single test set. The uncertainties are over different runs of the algorithm and not over different partitions of the data into training and test sets. I find this worrying -- is there a case (in these datasets that have been around for so long and so widely tested), there is a commmunity-wide hill climbing on the test set -- reporting results that just happen to be better than a previous attempt on the specific test set? Is it not time to pool all the data and do cross validation (by training and testing on different partitions) so that we can evaluate the uncertainty in these results more accurarately?

---

> ### Author Response · Authors · 2017-12-28
> **Response:**
>
> Thanks for your feedback. The paper was motivated by well-known facts from coding theorems, and thus our main interest was to investigate the statistical properties of representations. The writing, however, was completed in a haste, and we agree that we failed to identify a strong connection to the coding theorem in a persuasive way. Unfortunately, this became clear to us only after reading feedbacks from the three reviewers. We actually had several different options for summarizing our findings, and now it looks like the current writing was not a good choice and needs a major revision. Such a revision might not be adequate for what is allowed during this review process, and therefore we have made only minor revisions for now.
>
> >> but, if I am right, these are results quoted on a single test set.
> For MNIST, 5 tests were performed. For the others, single test was performed. We paid less attention to the number of tests, because our main point was not on performance improvement but on how a variety of regularizers (including a few that we have designed for the first time) behave in similar/different ways in terms of statistical properties while achieving a comparable or superior performance.

---

### Decision · Program_Chairs · 2018-01-29
**ICLR 2018 Conference Acceptance Decision**

**Decision:**

Reject

**Comment:**

The paper received scores of 5,5,5, with the reviewers agreeing the paper was marginally below the acceptance threshold. The main issue, raised by both R2 and R3 was that connection between representation learning in deep nets and coding theory was not fully justified/made.  With no reviewer advocating acceptance, it is not possible to accept the paper unfortunately.